# Opium as a risk factor for early-onset coronary artery disease: Results from the Milano-Iran (MIran) study

**Alberto Maino[1,2], Saeed Sadeghian[3]\*, Ilaria Mancini[4], Seyed Hesameddin Abbasi[3,5], Hamidreza Poorhosseini[3], Mohammad Ali Boroumand[3], Masoumeh Lotfi-Tokaldany[3], Arash Jalali[3], Maria Teresa Pagliari[2], Frits R. Rosendaal[6], Flora Peyvandi[2,4]**

**1** Unit of Internal Medicine, Azienda Provinciale per i Servizi Sanitari (APSS), Ospedale Santa Chiara, Trento, Italy, **2** Fondazione IRCCS Ca' Granda Ospedale Maggiore Policlinico, Angelo Bianchi Bonomi Hemophilia and Thrombosis Center, Milan, Italy, **3** Tehran Heart Center, Cardiovascular Disease Research Institute, Tehran University of Medical Sciences, Tehran, Iran, **4** Department of Pathophysiology and Transplantation, Università degli Studi di Milano, Milan, Italy, **5** Department of Global Health and Population, Bernard Lown Scholar in Cardiovascular Health, Harvard T.H. Chan School of Public Health, Boston, MA, United States of America, **6** Department of Clinical Epidemiology, Leiden University Medical Center, Leiden, The Netherlands

\* ssadeghian@sina.tums.ac.ir

**Data Availability Statement:** Data cannot be shared publicly due to Iranian regulations and ethics restrictions. Data might be available from the Luigi Villa Foundation (Milan, Italy), upon

## Abstract

The spreading of opium use poses new health related concerns. In some areas of Asia its use is believed to protect from cardiovascular disorders, such as coronary artery disease (CAD). However, whether opium use has an association with CAD is unclear. We aimed to investigate the association between non-medical opium use and CAD. We set up a case-control analysis, i.e., the Milano-Iran (MIran) study by enrolling consecutive young patients who underwent a coronary angiography at the Tehran Heart Center, between 2004 and 2011. Incident cases with CAD were contrasted with controls for opium use. Relative risks were calculated in terms of odds ratios (ORs) by logistic regression models adjusted for age, sex, cigarette smoking, body mass index, hypertension, hyperlipidaemia, and diabetes. Interaction analyses were performed between opium and major cardiovascular risk factors. 1011 patients with CAD (mean age 43.6 years) and 2002 controls (mean age 54.3 years) were included in the study. Habitual opium users had a 3.8-fold increased risk of CAD (95% CI 2.4–6.2) compared with non-users. The association was strongest for men, with a fully adjusted OR of 5.5 (95%CI 3.0–9.9). No interaction was observed for the combination of opium addiction and hypertension, or diabetes, but an excess in risk was found in opium users with hyperlipidaemia (OR 16.8, 95%CI 8.9–31.7, expected OR 12.2), suggesting supra-additive interaction. In conclusion, despite common beliefs, we showed that non-medical opium use is associated with an increased risk of CAD, even when other cardiovascular risk factors are taken into account.

reasonable request, and for researchers who meet the criteria for access to confidential data. Contact information: contact@fondazioneluigivilla.org, Tel: +39 02 55 10 709.

**Funding:** The study was partially supported by the Italian Ministry of Health (Bando Ricerca Corrente 2022) through the Fondazione IRCCS Ca' Granda Ospedale Maggiore Policlinico. Moreover, the Tehran Heart Center and the Università degli Studi di Milano provided logistical and staff support. The funders had no role in study design, data collection and analysis, decision to publish, or preparation of the manuscript."

**Competing interests:** The authors have declared that no competing interests exist.

## Introduction

Opium is a substance derived from the opium poppy plant, the derivatives of which are used for medical and non-medical purposes [1]. According to the last World Drug Report, the opium production worldwide continued its long upward trend by growing 7% year-on-year, driven by a dramatically increased demand for non-medical purposes. There were about 31 million opiates users worldwide in 2020, roughly 1.2% of the population aged 15 to 64 years, with more than half of the estimated number of annual opiate users residing in Asia [2]. Apart from being a social problem that could impose persistent and harmful effects on the total population, opium addiction is considered a personal risk [3]. Opium intake and dependence are detrimental to health and can result in injury and poor quality of life [4]. Current evidence suggests that opium consumption is associated with an increased risk of several comorbidities, including cancer [5, 6]. Nevertheless, a popular belief, even among some physicians in western and central Asia, is that long-term use of low dose opium has a protective effect against chronic diseases such as diabetes mellitus, hypertension and even atherosclerosis, with the potential to prolong survival by preventing cardiovascular disorders [7, 8].

Coronary artery disease (CAD) and ischaemic stroke, the most common and severe cardiovascular disorders worldwide, are the leading cause of death in western countries in both men and women. In the last decades a dramatic increase in their incidences in south Asian countries has been observed [9–11]. It has been estimated that cardiovascular disorder are responsible for about 46% of all death in Iran [12]. Despite the common belief and the general awareness about drug abuse, few studies have investigated the association between cardiovascular disorders and opium consumption. Their results are inconclusive for CAD and ischaemic stroke [13, 14]. However, there is evidence that opium consumption is associated with an increased risk of death from several causes, including cardiovascular diseases [15].

With this background, we set up a case-control analysis in the frame of the Milano–Iran (MIran) study to investigate the association between non-medical opium consumption and the risk of CAD in a young population.

## Patients and methods

### Patients

Cases belong to The Milano-Iran (MIran) study. The MIran study included young subjects (males <45 years, females <55 years) who underwent diagnostic coronary angiography (CAG) at the Tehran Heart Center (Iran) between June 2004 and July 2011, for the following indications: acute myocardial infarction (AMI), stable or unstable angina, atypical chest pain (with positive exercise test or myocardial nuclear scan), valvular heart disease candidate for catheterization, and subjects with peripheral vascular disease (aortic disease, renal artery stenosis, carotid artery stenosis) [16, 17].

For this study, consecutive patients with luminal stenosis greater than 50% in at least one main coronary artery or its branches at the CAG were included as cases (severe CAD). Patients with evidence of minimal CAD (luminal stenosis <50%) were excluded from the analysis, as well as patients with history of myocardial infarction.

Controls were recruited from the population of the Tehran Cohort Study [18]. The Tehran Cohort Study is a population-based cohort that has enrolled between 2016 and 2019 more than 8000 voluntary participants from the general population of Tehran city with the scope of detecting the risk factors for cardiovascular disease, trauma and their related psychological factors. To be included in the Tehran Cohort Study participants had to be more than 35 years old at the time of recruitment.

Subjects unrelated with cases and free from a history of CAD, AMI, symptoms of stable or unstable angina, coronary angioplasty or coronary artery bypass surgery, or known symptomatic valvular heart disease where randomly selected from the Tehran Cohort Study and frequency matched for sex.

The study was approved by the Ethical committee Board of Tehran Heart Center, and all patients and controls signed written informed consent before inclusion in the study (code of ethics for the Teheran Cohort Study IR.TUMS.MEDICINE.REC.1399.074). Additional information regarding the ethical, cultural, and scientific considerations specific to inclusivity in global research is included in the S1 File.

## Clinical variables

The information regarding demographic and clinical data was retrieved from the patients' medical records and from a confidential and detailed questionnaire filled by the participants. This included information regarding age, sex, ethnicity, and the presence of cardiovascular risk factors including body mass index (BMI), current or past tobacco use, and history of hypertension, hyperlipidaemia and diabetes. BMI was calculated as weight in kilograms (kg) divided by height in squared meters. Current smokers were those who smoked at least one cigarette per day, or who had stopped smoking for less than one year; former smokers were those who had stopped smoking for at least one year, and never smokers were those who have never smoked. Subjects were considered hypertensive in the presence of a blood pressure repeatedly higher that 140/90 mmHg or the use of antihypertensive medications. Hyperlipidaemia was defined as hypercholesterolemia (total cholesterol greater than 200 mg/dl) or hypertriglyceridemia (triglycerides greater than 200 mg/dl), or history of antihyperlipidemic drugs. Diabetes mellitus was considered in the presence of fasting blood glucose greater than 126 mg/dl or blood glucose greater than 200 mg/dl after oral glucose tolerance test, or in the presence of a history of diabetes mellitus, with or without the use of anti-diabetic medications. Therefore, no distinction was made between different types of diabetes mellitus.

Detailed information on the habits of opium consumption were retrieved. Patients were asked whether they were using or had ever used opium for recreational purpose, whether the use was occasional or habitual, the type of opium habitually consumed (teriak, sukhteh, and shireh), the pattern of assumption (oral or smoked), and the amount of intake in terms of grams per day. Teriak is the locally name for raw opium, directly obtained through ripening the poppy capsules. Teriak can be ingested or smoked with opium pipe. Sukhteh is the opium dross remaining after the opium is smoked, and can only be ingested. Shireh is the refined opium, obtained from boiling the opium dross in water, filtering the mixture several times, and then evaporating the filtrate. In can be taken orally or smoked. For the purpose of the present study, opium addiction was defined as using opium for more than one year habitually at the time of the enrolment or in the past. Therefore, the definition of opium addiction was self-reported in both cases and controls.

## Statistics

Mean and standard deviation were used to describe continuous variables, count and percentage for categorical ones. Multivariable logistic regression models were used to calculate odds ratios (OR) and corresponding 95% confidence intervals (CI) as measures of relative risk for the association between opium addiction and CAD. In model 1 analyses were adjusted only for age and sex (matching variable), in model 2 for cigarette smoking (being smoking also the major pattern of opium assumption), and in model 3 for BMI (included as continuous variable) and history of hypertension, dyslipidaemia and diabetes (potential confounders). Patients

with missing data were not included in model 2 and 3. The main analysis was stratified by sex, in order to investigate whether sex differences could have a role in the association between opium and CAD. In a secondary analysis, dummy variables were created to assess the combined effect on the risk of CAD of opium addiction and the presence of a major cardiovascular risk factor, such as hypertension, hyperlipidaemia and diabetes. Considering a prevalence of opium consumption in the Iranian population of 4%, we estimated that with a sample size of 1000 CAD cases and 2000 controls we were able to detect an OR of 1.7 or higher with 0.95 confidence level and a power of 0.8. Statistical analyses were performed by using the SPSS statistical software package (IBM SPSS Statistics for Windows, Version 22.0. Armonk, NY: IBM Corp). The manuscript follows the standards of the STrengthening the Reporting of OBservational studies in Epidemiology (STROBE) guidelines. The STROBE checklist is included in the S2 File.

## Results

One thousand and eleven patients with severe CAD and 2002 controls were included in the analysis. Demographic and clinical characteristics are summarized in Table 1. Cases were slightly younger than controls (mean age 46.6 vs 54.3), and among cases there was a greater prevalence of women (65.2% among cases and 56.7% among controls). As expected, known cardiovascular risk factors were more prevalent in cases than in controls. The prevalence of opium addiction in cases was 9.3% (94/1011), whereas in controls it was 3.3% (67/2002). Almost all opium-addicted subjects consumed only teriak opium, by smoking.

Opium addiction was associated with a 4-fold increased risk of CAD (OR 4.3, 95% CI 3.0–6.2), and this excess in risk decreased only slightly when all the major cardiovascular risk factors were taken into account (fully adjusted OR 3.8, 95% CI 2.4–6.2, Table 2). When analyses

**Table 1. Clinical characteristics and cardiovascular risk factors of incident cases with coronary artery disease (CAD) and controls.**

|  | CAD | controls |
|---|---|---|
|  | n = 1011 | n = 2002 |
| **Age (years, sd)** | 46.6 (5.8) | 54.3 (12.7) |
| **Age group (%)** |  |  |
| ≤39 years | 122 (12.1) | 274 (13.7) |
| 40–49 years | 501 (49.6) | 535 (26.7) |
| ≥50 years | 388 (38.3) | 1193 (59.6) |
| **Female (%)** | 659 (65.2) | 1135 (56.7) |
| **Smoking (%)** | 281 (27.8) | 329 (16.4) |
| **Hypertension (%)** | 566 (56.0) | 491 (24.5) |
| **Hyperlipidaemia (%)** | 758 (75.0) | 649 (32.4) |
| **Diabetes (%)** | 367 (36.3) | 336 (16.8) |
| **BMI (kg/m$^2$, sd)** | 29.7 (5.1) | 28.0 (5.0) |
| **BMI category (%)** |  |  |
| Normal weight (<25 kg/m$^2$) | 162 (16.1) | 559 (28.1) |
| Overweight (≥25 and <30 kg/m$^2$) | 416 (41.3) | 852 (42.9) |
| Obese (≥30 kg/m$^2$) | 430 (42.7) | 577 (29.0) |
| **Opium addiction (%)** | 94 (9.3) | 67 (3.3) |

Smoking includes both former and current cigarettes smokers, data available for 100% of cases and 98.9% of controls; BMI, body mass index, data available for 99.7% of cases and 99.3% of controls.

**Table 2. Risk of CAD by opium addiction.**

| Opium addiction | Cases n = 1011 | Controls n = 2002 | OR[1] (95%CI) | OR[2] (95%CI) | OR[3] (95%CI) |
|---|---|---|---|---|---|
| **All** | | | | | |
| no | 917 (90.7%) | 1935 (96.7%) | ref | ref | ref |
| yes | 94 (9.3%) | 67 (3.3%) | 4.3 (3.0–6.2) | 2.9 (2.0–4.3) | 3.8 (2.4–6.2) |
| **Male** | | | | | |
| no | 264 (75%) | 804 (92.7%) | ref | ref | ref |
| yes | 88 (25%) | 63 (7.3%) | 6.5 (4.1–10.3) | 3.8 (2.3–6.2) | 5.5 (3.0–9.9) |
| **Female** | | | | | |
| no | 653 (99.1%) | 1131 (99.6%) | ref | ref | ref |
| yes | 6 (0.9%) | 4 (0.4%) | 3.0 (0.8–10.9) | 2.5 (0.7–9.3) | 2.6 (0.5–12.6) |

Odds ratios are obtained by multivariate logistic regression. OR[1] are adjusted for age and sex; OR[2] are adjusted for OR[1] and additionally for smoking; OR[3] are adjusted for OR[2] and additionally for the presence of hypertension, dyslipidaemia, diabetes and BMI.

were stratified by sex, male opium users had more than 5-fold increased risk of CAD compared with male opium non-users (fully adjusted OR 5.5, 95% CI 3.0–9.9). This estimate was greater than the corresponding one in females (fully adjusted OR 2.6, 95% CI 0.5–12.6). However, among females, the overall prevalence of opium addiction was very low (10/1784, 0.6%).

Table 3 shows the combined effect of opium addiction and the presence of major cardiovascular risk factors. Hypertension was associated with an increased risk of CAD (OR 6.0, 95% CI 4.7–7.7). Opium addicted subjects without hypertension had a risk of CAD similar to that of the main analysis (OR 5.0, 95% CI 2.9–8.5). The risk of CAD conferred by the combination of opium addiction and hypertension was similar to the one expect by the sum of the two separate relative risks, making the presence of any biological interaction unlikely (OR for the combination 9.0, 95% CI 3.7–21.7; expected OR 1 + [6.0–1] + [5.0–1] = 10, being 1 the baseline risk). A small excess in risk was observed for the combination of opium addiction and

**Table 3. Risk of CAD in relation to the combination of opium addiction and the presence of the major cardiovascular risk factors.**

| Opium addiction | Hypertension | Cases n = 1011 | Controls n = 2002 | OR[1] (95%CI) | OR[2] (95%CI) |
|---|---|---|---|---|---|
| no | no | 380 (37.6%) | 1461 (73%) | ref | ref |
| no | yes | 537 (53.1%) | 474 (23.7%) | 10.4 (8.3–13.0) | 6.0 (4.7–7.7) |
| yes | no | 65 (6.4%) | 50 (2.5%) | 4.7 (2.9–7.5) | 5.0 (2.9–8.5) |
| yes | yes | 29 (2.9%) | 17 (0.8%) | 11.7 (5.3–25.9) | 9.0 (3.7–21.7) |
| **Opium addiction** | **Hyper-lipidaemia** | | | | |
| no | no | 218 (21.6%) | 1309 (65.4%) | ref | Ref |
| no | yes | 692 (69.1%) | 625 (31.2%) | 11.5 (9.3–14.3) | 7.2 (5.7–9.1) |
| yes | no | 43 (2.1%) | 34 (3.4%) | 5.6 (3.1–9.9) | 6.0 (3.2–11.3) |
| yes | yes | 60 (5.9%) | 24 (1.2%) | 21.4 (11.7–39.1) | 16.8 (8.9–31.7) |
| **Opium addiction** | **Diabetes** | | | | |
| no | no | 558 (55.2%) | 1613 (80.6%) | ref | ref |
| no | yes | 358 (35.4%) | 321 (16.0%) | 5.5 (4.4–6.8) | 2.8 (2.2–3.6) |
| yes | no | 85 (8.4%) | 52 (2.6%) | 3.7 (2.4–5.7) | 5.0 (3.0–8.4) |
| yes | yes | 9 (0.9%) | 15 (0.7%) | 3.9 (1.6–10.4) | 1.8 (0.5–5.9) |

Odds ratios are obtained by multivariate logistic regression. OR[1] are adjusted for age, sex and smoking; OR[2] are adjusted for OR[1] and BMI, and additionally for hypertension, dyslipidaemia and diabetes when indicated.

hyperlipidaemia. Subjects with hyperlipidaemia had an increased risk of CAD (OR 7.2, 95% CI 5.7–9.1), while the relative risk was high in hyperlipidaemic subjects with opium addiction (OR 16.8, 95% CI 8.9–31.7; expected OR 1 + [7.2–1] + [6.0–1] = 12.2), suggesting a positive interaction between opium and high serum lipids levels. On the contrary, diabetic patients with opium addiction had a lower risk of CAD than the sum of the two separate risk factors (OR 1.8, 95% CI 0.5–5.9; expected OR 1 + [2.8–1] + [5.0–1] = 6.8). However, in the latter analysis, numbers of patients with both risk factors were relatively small.

## Discussion

In a large young population from Iran, we investigated the association between non-medical opium consumption and the risk of early-onset CAD. We found that opium users were at an increased risk of CAD (OR 4.3, 95% CI 3.0–6.2) compared with non-users, even when other major cardiovascular risk factors were taken into account.

Very few studies have investigated the relationship between opium addiction and the risk of arterial thrombosis, with small sample size and with contrasting results, ranging from a protective to a harmful effect. Opium has been reported to be protective against the risk of ischaemic stroke, and not affecting the risk of ischaemic heart disease in a previous study [13]. However, Masoomi et al. suggested that opium use is associated with an increased risk of CAD in cases without cigarette smoking but not in addicted cases with cigarette smoking, by investigating a small population in Kerman, Iran (a case control study including 58 cases and 33 controls) [19]. Same conclusion was reported by Sadeghian et al., reporting unadjusted OR for the relationship between opium consumption and CAD in men of 4.5 (95% CI 1.5–13.4), very close to the ones found in our study [20]. In a previous study by Khademi et al., opium consumption has been associated with an increased risk of overall death in men and women, for major causes of death, including death from cardiovascular disease, and for different subtypes of opium and various routes of use [15]. The authors discussed whether this association was causal, or confounded by other shared risk factors, primarily the deleterious effects of smoking. Since smoking is the most used route of opium consumption (through smoking the absorption of morphine and codeine, the two sedative alkaloids that bind to the μ opioid receptor (MOR) in the brain, is quick across the lining of the lungs, and they achieve the target tissues within seconds), cigarette smoking is widespread among opium users. However, our results showed that the association between opium consumption and CAD decreased only slightly when smoking was taken into account (OR adjusted for smoking 2.9, 95% CI 2.0–4.3), and remained high even when other major cardiovascular risk factors were considered (fully adjusted OR 3.8, 95% CI 2.4–6.2). Our results from a large study confirm that opium addiction is a risk factor for CAD and suggest a causal relationship between the increased cardiovascular mortality found by Khademi et al., and opium consumption.

However, the biological mechanisms underlying this association are still to be clarified. It has been reported that opium users showed increased plasma levels of homocysteine (independently of the levels of vitamin $B_{12}$ or folate), fibrinogen and PAI-1 levels compared with non-users [21–24]. This might contributes to a hypercoagulable state that influences the atherosclerotic plaque formation and proliferation, as shown in preclinical studies on rats [25]. Other than a direct effect on the vessel wall, opium consumption might act through pathophysiological pathways related to atherosclerosis, such as hypertension, dyslipidaemia and diabetes, to which opium seems associated [8]. There is paucity of data with regard to the effects of opiates and opium on blood pressure. Acute or chronic exposure to morphine in rats significantly decreased the systolic, diastolic, and mean arterial blood pressures [26]. In humans, however, although opium might temporarily reduce blood pressure by vasodilation, no differences has

been found in long term diastolic and systolic blood pressure between opioid users and non-users [27, 28], even though an association with chronic kidney disease, which may affect the blood pressure, has been reported [29]. In our analysis, adjustment for hypertension did not change the results. Moreover, we found no interaction between hypertension and opium use in our population. Since the relative risk of CAD for the hypertensive subjects who used opium was closed to the expected one by the sum of the relative risk of hypertension and opium use, it seems that the two risk factors act on non-interacting pathways. On the contrary, in our study, patients with dyslipidaemia who used opium had an excess risk of CAD compared with dyslipidaemic patients who did not use opium (OR for the combination 16.8, 95% CI 8.9–31.7), suggesting a positive biological interaction between high lipids levels and opium. There is little evidence that opium use is associated with increased lipid levels in humans [28]. However, our observation is supported by animal models, in which opium consumption had worsening effects on atherosclerosis formation related with hypercholesterolemia, mainly affecting lipid profile [25]. Finally, the effect of opium on serum glucose levels is controversial. Studies have indicated that opium consumption is related with either an unaltered or a reduced serum glucose levels [30, 31]. In our study, diabetic patients who used opium seemed to be at a relatively lower risk of CAD compared with diabetic patients who did not use opium, suggesting a protective effect of opium on the risk of CAD in diabetic patients. However, this analysis included few subjects, and therefore, should be considered very cautiously.

Our study has several limitations. First, the use of opium was self-reported in our study. The prevalence of recreational opium consumption in the Iranian population has been reported to be as high as 14% in the Golestan cohort, much higher than the prevalence we have found within our controls subjects (3.3%) [32]. Part of this difference is related to the case-control design of our study, to the different study populations, and to the definition of opium users, that in our study included only regular users. However, although it has previously been demonstrated in an Iranian population that a self-reported use of opium can be a reliable measurement of its real consumption [33], an underestimation of the real prevalence of opium users in both cases and controls may have occurred. This might partly explain the low prevalence of opium consumption within women, especially when compared with men (0.4% in control women compared with 7.3% in control men). In our analysis, the association between opium and CAD was stronger for men (OR 5.5, 95% CI 3.0–9.9) than for women (OR 2.6, 95% CI 0.5–12.6). Because the number of exposed women in the latter analyses is too small, we cannot argue that this difference is related to a specific sex effect, or merely to chance. Differential misclassification is less likely to have occurred. Since there is no perception that opium might cause myocardial infarction, it is unlikely that an underestimation of the exposure occurred differently between cases and controls. Second, due to lack of data, we were unable to perform additional analyses on dose related response and route of consumption. Those analyses might have helped in understanding the biological mechanisms underlying the association. Third, although we adjusted the analysis for several factors, we cannot exclude that residual confounding, such as for example socioeconomic factors, personality trait, physical activity, associated substance abuse and medications, might have played a role in our results. Finally, despite our study is the largest on the relationship between opium and CAD, numbers in some sub-analyses were relatively small, leading to uncertainty and wide confidence intervals.

## Conclusions

In conclusion, we found that non-medical opium use is associated with an increased risk of CAD, even when several major cardiovascular risk factors are taken into account. The increased risk is consistent in both sexes and it is particularly high for subjects with

hyperlipidaemia. Our results should help implement prevention and educational programs against the non-medical use and spread of opium consumption worldwide, especially in Asian countries.

## Supporting information

**S1 File. Questionnaire on inclusivity in global research.**
(PDF)

**S2 File. STROBE checklist.**
(PDF)

## Author Contributions

**Conceptualization:** Alberto Maino, Seyed Hesameddin Abbasi, Frits R. Rosendaal, Flora Peyvandi.

**Data curation:** Saeed Sadeghian, Seyed Hesameddin Abbasi, Hamidreza Poorhosseini, Mohammad Ali Boroumand, Masoumeh Lotfi-Tokaldany, Arash Jalali.

**Formal analysis:** Alberto Maino, Ilaria Mancini, Maria Teresa Pagliari.

**Funding acquisition:** Flora Peyvandi.

**Methodology:** Alberto Maino, Seyed Hesameddin Abbasi, Frits R. Rosendaal.

**Project administration:** Alberto Maino.

**Supervision:** Flora Peyvandi.

**Validation:** Saeed Sadeghian.

**Writing – original draft:** Alberto Maino.

**Writing – review & editing:** Saeed Sadeghian, Ilaria Mancini, Seyed Hesameddin Abbasi, Hamidreza Poorhosseini, Mohammad Ali Boroumand, Masoumeh Lotfi-Tokaldany, Arash Jalali, Maria Teresa Pagliari, Frits R. Rosendaal, Flora Peyvandi.

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
