## [Decision Letter · Decision Letter 0]

13 Sep 2022

PONE-D-22-24696Recreational opium use as a risk factor for coronary artery disease: results from the premature coronary artery disease Milano-Iran (MIran) studyPLOS ONE

Dear Dr. Maino,

Thank you for submitting your manuscript to PLOS ONE. After careful consideration, we feel that it has merit but does not fully meet PLOS ONE’s publication criteria as it currently stands. Therefore, we invite you to submit a revised version of the manuscript that addresses the points raised during the review process.

A marked-up copy of your manuscript that highlights changes made to the original version. You should upload this as a separate file labeled 'Revised Manuscript with Track Changes'.An unmarked version of your revised paper without tracked changes. You should upload this as a separate file labeled 'Manuscript'.

We look forward to receiving your revised manuscript.

Kind regards,

Redoy Ranjan, MBBS, MRCSEd, Ch.M., MS (CV&TS), FACS

Academic Editor

PLOS ONE

Journal Requirements:

3. Please note that PLOS ONE has specific guidelines on code sharing for submissions in which author-generated code underpins the findings in the manuscript. In these cases, all author-generated code must be made available without restrictions upon publication of the work. Please review our guidelines at https://journals.plos.org/plosone/s/materials-and-software-sharing#loc-sharing-code and ensure that your code is shared in a way that follows best practice and facilitates reproducibility and reuse. New software must comply with the Open Source Definition.

4. Please include a complete copy of PLOS’ questionnaire on inclusivity in global research in your revised manuscript. Our policy for research in this area aims to improve transparency in the reporting of research performed outside of researchers’ own country or community. The policy applies to researchers who have travelled to a different country to conduct research, research with Indigenous populations or their lands, and research on cultural artefacts. The questionnaire can also be requested at the journal’s discretion for any other submissions, even if these conditions are not met.  Please find more information on the policy and a link to download a blank copy of the questionnaire here: https://journals.plos.org/plosone/s/best-practices-in-research-reporting. Please upload a completed version of your questionnaire as Supporting Information when you resubmit your manuscript.

"The Tehran Heart Center and the Università degli Studi di Milano provided logistical and staff support, without taking part in the design, data collection, analysis, interpretation and manuscript writing."

7. We note that you have stated that you will provide repository information for your data at acceptance. Should your manuscript be accepted for publication, we will hold it until you provide the relevant accession numbers or DOIs necessary to access your data. If you wish to make changes to your Data Availability statement, please describe these changes in your cover letter and we will update your Data Availability statement to reflect the information you provide.

8. Ethics statement appears in the Methods section of the manuscript AND at the end of the manuscript:

Your ethics statement should only appear in the Methods section of your manuscript. If your ethics statement is written in any section besides the Methods, please delete it from any other section. 

Reviewers' comments:

Reviewer's Responses to Questions

**Comments to the Author**

1. Is the manuscript technically sound, and do the data support the conclusions?

Reviewer #1: Partly

Reviewer #2: Partly

Reviewer #3: No

2. Has the statistical analysis been performed appropriately and rigorously? 

Reviewer #1: No

Reviewer #2: Yes

Reviewer #3: No

3. Have the authors made all data underlying the findings in their manuscript fully available?

Reviewer #1: No

Reviewer #2: Yes

Reviewer #3: Yes

4. Is the manuscript presented in an intelligible fashion and written in standard English?

Reviewer #1: Yes

Reviewer #2: Yes

Reviewer #3: Yes

5. Review Comments to the Author

**Reviewer #1:** Distinction between cases and controls’ criteria and the justification behind choosing such factors is not clear.

Effect modifiers like socioeconomic factors and personality trait etc. are not taken into account which might have skewed the result.

Must follow STROBE guideline, since this is a case-control study.

Also, variable selection for regression model using directed acyclic graph (DAG) analysis might avoid common pitfalls like treating a colloider variable as confounding variable and thus unintentionally introducing bias into the model.

**Reviewer #2:** I would like to congratulate you on this manuscript. However, you should try to make it better before it could be published

1. Please include the IRB number.

2. Most likely CAG instead of CA, please check the international approach to the write.

3. Authors should improve their style of scientific writeup throughout the manuscript. English language correction is deemed necessary.

Abstract:

Pose-poses

other cardiovascular risk- other cardiovascular risks

Introduction:

non-medical purpose- non-medical purposes

demand for non-medical purpose- demand for non-medical purposes

increase in their incidences- increase in their incidences

Patient:

belongs from- belong to

Clinical variables

data were- data was

and from a confidential- and from a confidential

one year habitually on- one year habitually at

For the purpose of the present study- For the purpose of the present study

Discussion:

various routs of use- various routes of use

Since the relative risk of CAD for hypertensive subject- Since the relative risk of CAD for the hypertensive subjects

Conclusion:

should help implementing- should help implement

A. Introduction:

1. In 2016, according to the last World Drug Report- Any new update on 2021 or 2022 report, if present added this

2. In this line “There were about 19.4 million opiates users worldwide in 66 2016, roughly 0.4% of the population aged 15 to 64 years, with more than half of the 67 estimated number of annual opiate users residing in Asia” compare with latest WHO report of worldwide and check any more current update

3. Give some relationship between homocysteine, Increased plasma fibrinogen, plasminogen activator Inhibitor-1 (PAI-1), and opium and CVD.

4. The number of cardiac patients in Iran and around the world since the introduction and description are both compared in this part.

B. Result:

1. To add how the current sample size was computed and details of the sampling procedure used and also the selection of control details.

2. To describe briefly how to select the variables for computing the logistic regression model.

3. How Confounders were identified?

4. Better to rework the result part.

C. In the discussion part:

1. Add- the cardiovascular impact of opium and potential mechanisms of action, Compare this study with other studies from Europe, the Middle East, and Asia.

2. More work have to do also in the discussion part.

Other comments:

1. Which Opium types are included (e.g: teriak, Sukhteh, and Shireh) all kindly give a slight description of this.

2. Any internal validation study has been performed in this population.

3. Any data for liver function tests that are available, if any, should be compared in the analysis.

4. The age group, BMI, and other relevant group relationships should be classified, if at all possible.

**Reviewer #3:** Many studies have already found that opium use significantly increase cardiovascular events. The authors of the study featured an association of its use and CAD.

The reviewer found that there are structural problems.

Specific comments:

1. The analysis was performed after confirmation of CAD. Therefore, the study cannot show the relationship between CAD and opium intake.

If the authors want to show that opium use might be a risk factor for CAD, they should prospectively collect data on differences in incidence of CAD between no-CAD subjects with and without opium intake,

2. The authors described that opium use is believed to protect from cardiovascular disorders in some areas of Asia. Please add references.

Ref 7 (Gen Thorac Cardiovasc Surg. 2010;58:62) suggests that continued use of opium significantly predicts re-hospitalization with a cardiac cause after CABG surgery but not that opium use prevents CV events.

3. There are some spelling mistakes in the text (ex. line 192 ‘ischaemic hearth disease’).

6. PLOS authors have the option to publish the peer review history of their article (what does this mean?). If published, this will include your full peer review and any attached files.

Reviewer #1: No

Reviewer #2: No

Reviewer #3: **Yes: **Hideki ISHII

---

## [Author Response · Author response to Decision Letter 0]

20 Nov 2022

Response to Reviewers

We would like to thank the reviewers for their useful and constructive comments. We revised the manuscript according to their comments, and we believe that the report has improved in both clarity and level of detail. 

Please find below our point-by-point response.

 

Reviewers Comments

Reviewer #1

- Distinction between cases and controls’ criteria and the justification behind choosing such factors is not clear.

Cases were young subjects with symptomatic severe coronary artery disease (CAD) detected by coronary angiography and with no history of previous myocardial infarction. Controls were subjects unrelated with cases and free from a history of CAD, AMI, symptoms of stable or unstable angina, known symptomatic valvular heart disease, coronary angioplasty or coronary artery bypass surgery.

 We realized that the description of cases and controls might be unclear. Therefore, we changed the text in the methods section and added references to the following two previously published papers: (for cases) Abbasi SH et al., Introducing the Tehran Heart Center's Premature Coronary Atherosclerosis Cohort: THC-PAC Study. J Tehran Heart Cent. 2015;10(1):34-42. PMID: 26157461; (for controls) Akbar Shafiee et al., Tehran cohort study (TeCS) on cardiovascular diseases, injury, and mental health: Design, methods, and recruitment data. Global Epidemiology Volume 3, 2021, 100051, doi: 10.1016/j.gloepi.2021.100051.

Thanks to the reviewer comment, we have also realized that the enrolment period of the patient cohort was incorrect in the present manuscript. Therefore, the text has been revised with the correct dates (June 2004-July 2011) [see Abbasi et al.].

The Methods section at page 5, line 82, now reads as follow:

“Cases belong to The Milano-Iran (MIran) study. The MIran study included young subjects (males <45 years, females <55 years) who underwent diagnostic coronary angiography (CAG) at the Tehran Heart Center (Iran) between June 2004 and July, for the following indications: acute myocardial infarction (AMI), stable or unstable angina, atypical chest pain (with positive exercise test or myocardial nuclear scan), valvular heart disease candidate for catheterization, and subjects with peripheral vascular disease (aortic disease, renal artery stenosis, carotid artery stenosis).[16,17] 

For this study, consecutive patients with luminal stenosis greater than 50% in at least one main coronary artery or its branches at the CAG were included as cases (severe CAD). Patients with evidence of minimal CAD (luminal stenosis <50%) were excluded from the analysis, as well as patients with history of myocardial infarction. 

Controls were recruited from the population of the Tehran Cohort Study.[18] The Tehran Cohort Study is a population-based cohort that has enrolled between 2016 and 2019 more than 8000 voluntary participants from the general population of Tehran city with the scope of detecting the risk factors for cardiovascular disease, trauma and their related psychological factors. To be included in the Tehran Cohort Study participants should be more than 35 years old at the time of recruitment. 

Subjects unrelated with cases and free from a history of CAD, AMI, symptoms of stable or unstable angina, coronary angioplasty or coronary artery bypass surgery, or known symptomatic valvular heart disease where randomly selected from the Tehran Cohort Study and frequency matched for sex. 

The study was approved by the Ethical committee Board of Tehran Heart Center, and all patients and controls signed written informed consent before inclusion in the study.“

- Effect modifiers like socioeconomic factors and personality trait etc. are not taken into account which might have skewed the result.

We totally agree with the reviewer. Unfortunately, we do not have such information in our study. We stressed this point in the discussion section under limitations, as follow (page 15, line 288)

“Third, although we adjusted the analysis for several factors, we cannot exclude that residual confounding, such as for example socioeconomic factors, personality trait, physical activity, associated substance abuse and medications, might have played a role in our results.”

- Must follow STROBE guideline, since this is a case-control study.

We made all our effort to adhere to the STROBE guidelines. In the revised version of the manuscript, we checked whether all the items of the STROBE checklist were met and, where needed, we implemented the manuscript accordingly (we added a sentence on the management of missing data and calculation of the sample size). We now included, as a supplemental file, the completed STROBE checklist.

- Also, variable selection for regression model using directed acyclic graph (DAG) analysis might avoid common pitfalls like treating a colloider variable as confounding variable and thus unintentionally introducing bias into the model.

We thank the reviewer for this important comment. During the study plan we reasoned about which variable had to be included in the regression model. The ones we chose were the major risk factor for CAD (such as BMI, hypertension, diabetes, dyslipidaemia and cigarette smoking). No one of these can be considered a collider. Moreover, to better understand the association between opium consumption and CAD we showed different models with increasing number of covariates. The point estimates of all these models were similar to the crude estimate, indicating that no one of such variable is a collider, neither a substantial confounder.

 

Reviewer #2: I would like to congratulate you on this manuscript. However, you should try to make it better before it could be published

- 1. Please include the IRB number. 

The board of research and the committee of medical ethics at Tehran University of Medical Sciences approved the protocol of the Teheran Cohort Study with a code of ethics IR.TUMS.MEDICINE.REC.1399.074. 

Page 6, line 112

‘The study was approved by the Ethical committee Board of Tehran Heart Center, and all patients and controls signed written informed consent before inclusion in the study (code of ethics for the Teheran Cohort Study IR.TUMS.MEDICINE.REC.1399.074).’

- 2. Most likely CAG instead of CA, please check the international approach to the write.

We thank the reviewer for this clarification. After careful check, we realized that CAG is more used than CA for coronary angiogram. We therefore changed the acronym throughout the text.

- 3. Authors should improve their style of scientific writeup throughout the manuscript. English language correction is deemed necessary.

Abstract:

Pose-poses

other cardiovascular risk- other cardiovascular risks

Introduction:

non-medical purpose- non-medical purposes

demand for non-medical purpose- demand for non-medical purposes

increase in their incidences- increase in their incidences

Patient:

belongs from- belong to

Clinical variables

data were- data was

and from a confidential- and from a confidential

one year habitually on- one year habitually at

For the purpose of the present study- For the purpose of the present study

Discussion:

various routs of use- various routes of use

Since the relative risk of CAD for hypertensive subject- Since the relative risk of CAD for the hypertensive subjects

Conclusion:

should help implementing- should help implement

We thank the reviewer for having read the manuscript so carefully. We revised the text and we addressed all the corrections suggested.

- A. Introduction:

1. In 2016, according to the last World Drug Report- Any new update on 2021 or 2022 report, if present added this

Thank you for this comment. We updated the data with the updated World Drug Report, released on June 2022. The introduction now read as follow (page 3, line 53): 

“According to the last World Drug Report, the opium production worldwide continued its long upward trend by growing 7% year-on-year, driven by a dramatically increased demand for non-medical purposes. There were about 31 million opiates users worldwide in 2020, roughly 1.2% of the population aged 15 to 64 years, with more than half of the estimated number of annual opiate users residing in Asia.[2]”

- 2. In this line “There were about 19.4 million opiates users worldwide in 66 2016, roughly 0.4% of the population aged 15 to 64 years, with more than half of the 67 estimated number of annual opiate users residing in Asia” compare with latest WHO report of worldwide and check any more current update

Please, refer to the answer to the previous comment. 

- 3. Give some relationship between homocysteine, Increased plasma fibrinogen, plasminogen activator Inhibitor-1 (PAI-1), and opium and CVD.

We thank the reviewer for this suggestion. We have now included in the discussion a comment on the possible implication of homocysteine, plasma fibrinogen and PAI in the relationship between opium use and CAD. The new comment in the discussion reads as follow (page 13, line 237):

“However, the biological mechanisms underlying this association are still to be clarified. It has been reported that opium users showed increased plasma levels of homocysteine (independently of the levels of vitamin B12 or folate), fibrinogen and PAI-1 levels compared with non-users.[17–20] This might contributes to a hypercoagulable state that influences the atherosclerotic plaque formation and proliferation, as shown in preclinical studies on rats.[21]”

- 4. The number of cardiac patients in Iran and around the world since the introduction and description are both compared in this part.

We added information available in literature about cardiovascular disorders in Iran (page 3, line 67)

“Coronary artery disease (CAD) and ischaemic stroke, the most common and severe cardiovascular disorders worldwide, are the leading cause of death in western countries in both men and women. In the last decades a dramatic increase in their incidences in south Asian countries has been observed.[9–11] It has been estimated that cardiovascular disorder are x\\ 46% of all death in Iran.[12]”

- B. Result:

1. To add how the current sample size was computed and details of the sampling procedure used and also the selection of control details.

We added in the method section a sentence about the sample size calculation, which reads as follow (page 7, line 146):

“Considering a prevalence of opium consumption in the Iranian population of 0.04, we estimated that with a sample size of 1000 CAD cases and 1000 controls we were able to detect an OR of 1.7 or higher with 0.95 confidence level and a power of 0.8. “ 

Moreover, we realized that the description of cases and controls might be unclear. Therefore, we changed the text in the methods section (page 5, line 82), now reading as follow:

“Cases belong to The Milano-Iran (MIran) study. The MIran study included young subjects (males <45 years, females <55 years) who underwent diagnostic coronary angiography (CAG) at the Tehran Heart Center (Iran) between June 2004 and July, for the following indications: acute myocardial infarction (AMI), stable or unstable angina, atypical chest pain (with positive exercise test or myocardial nuclear scan), valvular heart disease candidate for catheterization, and subjects with peripheral vascular disease (aortic disease, renal artery stenosis, carotid artery stenosis).[16,17] 

For this study, consecutive patients with luminal stenosis greater than 50% in at least one main coronary artery or its branches at the CAG were included as cases (severe CAD). Patients with evidence of minimal CAD (luminal stenosis <50%) were excluded from the analysis, as well as patients with history of myocardial infarction. 

Controls were recruited from the population of the Tehran Cohort Study.[18] The Tehran Cohort Study is a population-based cohort that has enrolled between 2016 and 2019 more than 8000 voluntary participants from the general population of Tehran city with the scope of detecting the risk factors for cardiovascular disease, trauma and their related psychological factors. To be included in the Tehran Cohort Study participants should be more than 35 years old at the time of recruitment. 

Subjects unrelated with cases and free from a history of CAD, AMI, symptoms of stable or unstable angina, coronary angioplasty or coronary artery bypass surgery, or known symptomatic valvular heart disease where randomly selected from the Tehran Cohort Study and frequency matched for sex. 

The study was approved by the Ethical committee Board of Tehran Heart Center, and all patients and controls signed written informed consent before inclusion in the study. Additional information regarding the ethical, cultural, and scientific considerations specific to inclusivity in global research is included in the Supporting Information (S1 file).”

- 2. To describe briefly how to select the variables for computing the logistic regression model.

The logistic regression model includes the matching variable (sex), and any variable that we believed, a priori, to be a potential confounder in the association between opium consumption and CAD, such as cigarette smoking, BMI, hypertension, dyslipidaemia and diabetes. The sentence in methods now reads as follow (page 7, line 138).

“In model 1 analyses were adjusted only for age and sex (matching variable), in model 2 for cigarette smoking, and in model 3 for BMI (included as continuous variable) and history of hypertension, dyslipidaemia and diabetes (potential confounders). The main analysis was stratified by sex, in order to investigate whether sex differences could have a role in the association between opium and CAD. In a secondary analysis, dummy variables were created to assess the combined effect on the risk of CAD of opium addiction and the presence of a major cardiovascular risk factor, such as hypertension, hyperlipidaemia and diabetes.”

- 3. How Confounders were identified?

During the study plan we reasoned about which variable had to be included in the regression model as a potential confounder. The ones we chose were the major risk factor for CAD (such as BMI, hypertension, diabetes, dyslipidaemia and cigarette smoking). Moreover, to better understand the association between opium consumption and CAD, we showed different models with increasing number of covariates. 

- 4. Better to rework the result part.

Please, see the responses to the comments above. 

- C. In the discussion part:

1. Add- the cardiovascular impact of opium and potential mechanisms of action, Compare this study with other studies from Europe, the Middle East, and Asia.

We thank the reviewer for this comment. We rewrote the discussion section in the part about the possible pathophysiological effect of opium on the cardiovascular system (page 13, line 237). 

“However, the biological mechanisms underlying this association are still to be clarified. It has been reported that opium users showed increased plasma levels of homocysteine (independently of the levels of vitamin B12 or folate), fibrinogen and PAI-1 levels compared with non-users.[21–24] This might contributes to a hypercoagulable state that influences the atherosclerotic plaque formation and proliferation, as shown in preclinical studies on rats.[25] Other than a direct effect on the vessel wall, opium consumption might act through pathophysiological pathways related to atherosclerosis, such as hypertension, dyslipidaemia and diabetes, to which opium seems associated.[8] There is paucity of data with regard to the effects of opiates and opium on blood pressure. Acute or chronic exposure to morphine in rats significantly decreased the systolic, diastolic, and mean arterial blood pressures.[26] In humans, however, although opium might temporarily reduce blood pressure by vasodilation, no differences has been found in long term diastolic and systolic blood pressure between opioid users and non-users[27,28], even though an association with chronic kidney disease, which may affect the blood pressure, has been reported.[29] In our analysis, adjustment for hypertension did not change the results. Moreover, we found no interaction between hypertension and opium use in our population. Since the relative risk of CAD for the hypertensive subjects who used opium was closed to the expected one by the sum of the relative risk of hypertension and opium use, it seems that the two risk factors act on non-interacting pathways. On the contrary, in our study, patients with dyslipidaemia who used opium had an excess risk of CAD compared with dyslipidaemic patients who did not use opium (OR for the combination 16.8, 95% CI 8.9 - 31.7), suggesting a positive biological interaction between high lipids levels and opium. There is little evidence that opium use is associated with increased lipid levels in humans.[28] However, our observation is supported by animal models, in which opium consumption had worsening effects on atherosclerosis formation related with hypercholesterolemia, mainly affecting lipid profile.[25] Finally, the effect of opium on serum glucose levels is controversial. Studies have indicated that opium consumption is related with either an unaltered or a reduced serum glucose levels.[30,31] In our study, diabetic patients who used opium seemed to be at a relatively lower risk of CAD compared with diabetic patients who did not use opium, suggesting a protective effect of opium on the risk of CAD in diabetic patients. However, this analysis included few subjects, and therefore, should be considered very cautiously.”

- 2. More work have to do also in the discussion part.

Please see the answers to the previous comments.

- Other comments:

1. Which Opium types are included (e.g: teriak, Sukhteh, and Shireh) all kindly give a slight description of this.

The opium included in the study is teriak and was smoked by the 89% of opium-addicted. We added a description of opium types in the text. 

Page 7 line 126

‘Detailed information on the habits of opium consumption were retrieved. Patients were asked whether they were using or had ever used opium for recreational purpose, whether the use was occasional or habitual, the type of opium habitually consumed (teriak, sukhteh, and shireh), the pattern of assumption (oral or smoked), and the amount of intake in terms of grams per day. Teriak is the locally name for raw opium, directly obtained through ripening the poppy capsules. Teriak can be ingested or smoked with opium pipe. Sukhteh is the opium dross remaining after the opium is smoked, and can only be ingested. Shireh is the refined opium, obtained from boiling the opium dross in water, filtering the mixture several times, and then evaporating the filtrate. In can be taken orally or smoked.’

Page 9, line 166

‘Almost all opium-addicted subjects consumed only teriak opium, by smoking.’

- 2. Any internal validation study has been performed in this population.

We do not have a formal internal validation study. However, the prevalence of opium addiction in our control group is similar to what reported in another Iranian cohort, the Kerman Coronary Artery Disease Risk Factors Study. Indeed, the prevalence of opium addiction in our control group of subjects younger than 45 years old is 3.3%; the prevalence of opium addiction in the Kerman study ranges from 2.1% in the 24-35 age group to 4.1% in the 35-44 age group.

Moreover, in our study cases and controls were selected from a population of two cohorts whose description has been already published, as included in the new version of the manuscript: (for cases) Abbasi SH et al., Introducing the Tehran Heart Center's Premature Coronary Atherosclerosis Cohort: THC-PAC Study. J Tehran Heart Cent. 2015;10(1):34-42. PMID: 26157461; (for controls) Akbar Shafiee et al., Tehran cohort study (TeCS) on cardiovascular diseases, injury, and mental health: Design, methods, and recruitment data. Global Epidemiology Volume 3, 2021, 100051, doi: 10.1016/j.gloepi.2021.100051.

- 3. Any data for liver function tests that are available, if any, should be compared in the analysis.

Unfortunately, no data on liver function tests are available in the MIran.

- 4. The age group, BMI, and other relevant group relationships should be classified, if at all possible.

Age groups and BMI categories have been added in Table 1 of the new version of the manuscript (Page 10, line 163).

‘Table 1. Clinical characteristics and cardiovascular risk factors of incident cases with coronary artery disease (CAD) and controls.’

 CAD

n=1011 controls

n=2002

Age (years, sd) 46.6 (5.8) 54.3 (12.7)

Age group (%) 

≤39 years 122 (12.1) 274 (13.7)

40-49 years 501 (49.6) 535 (26.7)

≥50 years 388 (38.3) 1193 (59.6)

Female (%) 659 (65.2) 1135 (56.7)

Smoking (%) 281 (27.8) 329 (16.4)

Hypertension (%) 566 (56.0) 491 (24.5)

Hyperlipidaemia (%) 758 (75.0) 649 (32.4)

Diabetes (%) 367 (36.3) 336 (16.8)

BMI (kg/m2, sd) 29.7 (5.1) 28.0 (5.0)

BMI category (%) 

Normal weight (<25 kg/m2) 162 (16.1) 559 (28.1)

Overweight (≥25 and <30 kg/m2) 416 (41.3) 852 (42.9)

Obese (≥30 kg/m2) 430 (42.7) 577 (29.0)

Opium addiction (%) 94 (9.3) 67 (3.3)

 

Reviewer #3: Many studies have already found that opium use significantly increase cardiovascular events. The authors of the study featured an association of its use and CAD.

The reviewer found that there are structural problems.

Specific comments:

- 1. The analysis was performed after confirmation of CAD. Therefore, the study cannot show the relationship between CAD and opium intake.

If the authors want to show that opium use might be a risk factor for CAD, they should prospectively collect data on differences in incidence of CAD between no-CAD subjects with and without opium intake,

We thank the reviewer for their important comments. However, regarding to this point, we do not agree. 

We set up a case-control study comparing CAD cases with non-CAD controls. The case-control family of study designs is an important tool for identifying causal relations and frequently used in medical research. These study designs are particularly attractive for non-common diseases, as they are able to sample known cases of disease, vs. following a large number of subjects and waiting for disease onset in a relatively small number of individuals. Because confounding is the most well-known source of bias affecting causal inference from observational data, we made all the efforts to prevent any source of bias when we designed the study, and when we analysed the data, by adjusting for potential confounders that might affect the relationship between opium consumption and CAD.

In our analysis, opium consumption is associated with CAD even when several potential confounding and other atherothrombotic risk factors, such as diabetes, hypertension, dyslipidaemia and tobacco smoking, are taken into account. Therefore, we can conclude that opium consumption might be a risk factor for CAD.

- Rothman KJ, Greenland S, Lash TL. Modern Epidemiology, Third edition. Philadelphia: Lippincott Williams & Wilkins; 2008.

- L. Penning de Vries, B.B., Groenwold, R.H.H. Identification of causal effects in case-control studies. BMC Med Res Methodol 22, 7 (2022). https://doi.org/10.1186/s12874-021-01484-7

- 2. The authors described that opium use is believed to protect from cardiovascular disorders in some areas of Asia. Please add references.

Ref 7 (Gen Thorac Cardiovasc Surg. 2010;58:62) suggests that continued use of opium significantly predicts re-hospitalization with a cardiac cause after CABG surgery but not that opium use prevents CV events.

Thank you for this comment. We realized the mistake and we modified the reference accordingly. We now cite:

- Marmor M, Penn A, Widmer K, Levin RI, Maslansky R. Coronary artery disease and opioid use. Am J Cardiol. 2004;93(10):1295–7. 

- Najafipour H, Masoumi M, Amirzadeh R, Rostamzadeh F, Foad R, Farrokhi MS. Trends in the Prevalence and Incidence of Opium Abuse and its Association with Coronary Artery Risk Factors in Adult Population in Iran: Findings from Kerman Coronary Artery Disease Risk Factors Study. Iran J Med Sci. 2022;47(4):328–37. 

- 3. There are some spelling mistakes in the text (ex. line 192 ‘ischaemic hearth disease’).

Thank you for this comment, raised also by reviewer n° 2. We revised the text and we addressed the suggested corrections.

---

## [Decision Letter · Decision Letter 1]

11 Jan 2023

PONE-D-22-24696R1Recreational opium use as a risk factor for coronary artery disease: results from the premature coronary artery disease Milano-Iran (MIran) studyPLOS ONE

Dear Dr. Maino,

Thank you for submitting your manuscript to PLOS ONE. After careful consideration, we feel that it has merit but does not fully meet PLOS ONE’s publication criteria as it currently stands. Therefore, we invite you to submit a revised version of the manuscript that addresses the points raised during the review process.

ACADEMIC EDITOR: The authors are thanked for this submission to PLOS ONE. After a critical external peer review by three experts, I reinforce improving the clarity and presentation of your paper. Please see the attached reviewer comments detail below.

A rebuttal letter that responds to each point raised by the academic editor and reviewer(s). You should upload this letter as a separate file labeled 'Response to Reviewers'. A marked-up copy of your manuscript that highlights changes made to the original version. You should upload this as a separate file labeled 'Revised Manuscript with Track Changes'. An unmarked version of your revised paper without tracked changes. You should upload this as a separate file labeled 'Manuscript'. 

We look forward to receiving your revised manuscript.

Kind regards,

Redoy Ranjan, MBBS, MRCSEd, Ch.M., MS (CV&TS), FACS

Academic Editor

PLOS ONE

Reviewers' comments:

Reviewer's Responses to Questions

**Comments to the Author**

1. If the authors have adequately addressed your comments raised in a previous round of review and you feel that this manuscript is now acceptable for publication, you may indicate that here to bypass the “Comments to the Author” section, enter your conflict of interest statement in the “Confidential to Editor” section, and submit your "Accept" recommendation.

Reviewer #1: All comments have been addressed

Reviewer #2: All comments have been addressed

Reviewer #4: (No Response)

2. Is the manuscript technically sound, and do the data support the conclusions?

Reviewer #1: Yes

Reviewer #2: Yes

Reviewer #4: No

3. Has the statistical analysis been performed appropriately and rigorously? 

Reviewer #1: Yes

Reviewer #2: Yes

Reviewer #4: I Don't Know

4. Have the authors made all data underlying the findings in their manuscript fully available?

Reviewer #1: Yes

Reviewer #2: Yes

Reviewer #4: Yes

5. Is the manuscript presented in an intelligible fashion and written in standard English?

Reviewer #1: Yes

Reviewer #2: Yes

Reviewer #4: Yes

6. Review Comments to the Author

Reviewer #1: (No Response)

Reviewer #2: (No Response)

Reviewer #4: This revised version has substantial improvements according to previous comments.

I can make a number of specific comments:

- Considering that opium, at least in Latin America, is considered a serious drug, in my opinion the term recreational for this kind of drug is dangerous and should be used.

- I also contend the use of the term “premature”, since invasive detection of >50% stenosis does not represent premature disease.

- I don’t understand the need for discrimination between logistic regression models 2 and 3. Indeed, regarding the models created, I suggest revision by an expert. Please provide a table with the results from the multivariate analysis using the different models and the HR of each variable within the model.

- If you calculated a sample size for the control group of 1000 patients. Why did you include 2002?

- In my opinion, the design of the study is somewhat odd, since you cannot be sure whether the subjects within the control group are free from CAD. Aside from that, why didn’t you match age and sex between groups? P values or confidence intervals should be provided for Table 1.

- I believe that given such limitations regarding comparisons between groups, it might have been more appropriate using other design, maybe including only patients with suspected CAD within the MIran cohort and discriminating them according to extension indexes such as Leaman or Duke CAD indexes; or even evaluating opium use among patients with acute vs. stable coronary syndromes.

- Please explain why the prevalence of opium consumption was so much higher than expected in the power calculation.

7. PLOS authors have the option to publish the peer review history of their article (what does this mean?). If published, this will include your full peer review and any attached files.

Reviewer #1: **Yes: **Saumitra Chakravarty

Reviewer #2: **Yes: **Mohammad Ashraful Amin

Reviewer #4: **Yes: **Gaston Rodriguez-Granillo

---

## [Author Response · Author response to Decision Letter 1]

6 Mar 2023

Dear editor and reviewers, 

please find our point by point response in the attached files. 

Best regards, 

Alberto Maino

---

## [Decision Letter · Decision Letter 2]

14 Mar 2023

Opium as a risk factor for early-onset coronary artery disease: results from the Milano-Iran (MIran) study

PONE-D-22-24696R2

Dear Dr. Maino,

We’re pleased to inform you that your manuscript has been judged scientifically suitable for publication and will be formally accepted for publication once it meets all outstanding technical requirements.

Kind regards,

Redoy Ranjan, MBBS, MRCSEd, Ch.M., MS (CV&TS), FACS

Academic Editor

PLOS ONE

Additional Editor Comments (optional): The authors are thanked for this submission to PLOS ONE. After a critical external peer review by the experts and considering the overall reviewers' comments and authors' responses, your manuscript meets PLOS ONE's publication criteria; fulfils the methodological rigour and ethical standards.

**Reviewer's Responses to Questions**

1. If the authors have adequately addressed your comments raised in a previous round of review and you feel that this manuscript is now acceptable for publication, you may indicate that here to bypass the “Comments to the Author” section, enter your conflict of interest statement in the “Confidential to Editor” section, and submit your "Accept" recommendation.

Reviewer #1: All comments have been addressed

Reviewer #2: All comments have been addressed

---

## [Editor Report · Acceptance letter]

22 Mar 2023

PONE-D-22-24696R2 

Opium as a risk factor for early-onset coronary artery disease: results from the Milano-Iran (MIran) study 

Dear Dr. Maino:

I'm pleased to inform you that your manuscript has been deemed suitable for publication in PLOS ONE. Congratulations! Your manuscript is now with our production department. 

Kind regards, 

on behalf of

Dr. Redoy Ranjan 

Academic Editor

PLOS ONE